# The PPIP5K Family Member Asp1 Controls Inorganic Polyphosphate Metabolism in *S. pombe*

**DOI:** 10.3390/jof7080626

**Published:** 2021-07-31

**Authors:** Marina Pascual-Ortiz, Eva Walla, Ursula Fleig, Adolfo Saiardi

**Affiliations:** 1Eukaryotische Mikrobiologie, Institut für Funktionelle Genomforschung der Mikroorganismen, Heinrich Heine University, 40225 Düsseldorf, Germany; marina.pascual@uchceu.es (M.P.-O.); eva.walla@hhu.de (E.W.); 2Department of Biomedical Sciences, Faculty of Health Sciences, Universidad Cardenal Herrera, CEU Universities, 46115 Valencia, Spain; 3Medical Research Council Laboratory for Molecular Cell Biology, University College London, London WC1E 6BT, UK

**Keywords:** inorganic polyphosphate, phosphate homeostasis, inositol pyrophosphates, *Schizosaccharomyces pombe*, PPIP5K, Asp1, yeast

## Abstract

Inorganic polyphosphate (polyP) which is ubiquitously present in both prokaryotic and eukaryotic cells, consists of up to hundreds of orthophosphate residues linked by phosphoanhydride bonds. The biological role of this polymer is manifold and diverse and in fungi ranges from cell cycle control, phosphate homeostasis and virulence to post-translational protein modification. Control of polyP metabolism has been studied extensively in the budding yeast *Saccharomyces cerevisiae.* In this yeast, a specific class of inositol pyrophosphates (IPPs), named IP_7_, made by the IP6K family member Kcs1 regulate polyP synthesis by associating with the SPX domains of the vacuolar transporter chaperone (VTC) complex. To assess if this type of regulation was evolutionarily conserved, we determined the elements regulating polyP generation in the distantly related fission yeast *Schizosaccharomyces pombe*. Here, the VTC machinery is also essential for polyP generation. However, and in contrast to *S. cerevisiae*, a different IPP class generated by the bifunctional PPIP5K family member Asp1 control polyP metabolism. The analysis of Asp1 variant *S. pombe* strains revealed that cellular polyP levels directly correlate with Asp1-made IP_8_ levels, demonstrating a dose-dependent regulation. Thus, while the mechanism of polyP synthesis in yeasts is conserved, the IPP player regulating polyP metabolism is diverse.

## 1. Introduction

Inorganic polyphosphate (polyP), the linear polymer of phosphate residues, is ubiquitously present in living organisms. Although polyP is one of the simplest of biological polymers, it regulates a pleiotropy of biological processes. Intrinsic to its polymeric charged nature is its ability to regulate phosphate and cation homeostasis. Additionally, polyP controls bacterial virulence [1], energy metabolism [2], protein folding [3], cell cycle progression [4], and signal transduction via lysine-polyphosphorylation [5]. This list of polyP-modulated functions is not exhaustive, and we recommend the interested readers the following polyP reviews [6,7,8,9].

The enzymology of synthesis of polyP is known for bacteria and for a number of unicellular eukaryotes, while in metazoans, the synthesis of polyP remains an open question [10]. Bacteria have two types of enzymes capable of synthesizing polyP, named polyphosphatekinases PPK1 and PPK2 [11]. In the social amoeba *Dictyostelium discoideum*, polyP is synthesized via a PPK1 homologous gene that was acquired by horizontal gene transfer [12]. In fungi such as the budding yeast *Saccharomyces cerevisiae* and protists such as Trypanosoma and Leishmania, the vacuole transporter chaperone (VTC) complex is responsible for polyP synthesis [13,14]. This complex is best characterized in *S. cerevisiae* were the VTC subunit four (Vtc4) possesses the catalytic domain synthesizing polyP [15]. Consequently, yeast strains with a *VTC4* deletion have no or extremely low levels of polyP [16].

In *S. cerevisiae* polyP synthesis is regulated by inositol pyrophosphates (IPPs also known as PP-IP) that are a class of cellular messengers possessing one or more phosphoanhydride (pyro) moiety/ies. IPP synthesis is mediated by a cascade of conserved enzymes, starting with phospholipase C hydrolyzing PI(4,5)P_2_ to IP_3_ and diacylglycerol [17]. IP_3_ is the substrate of the Arg82 inositol polyphosphate multikinase generating IP_4_ and IP_5_ [18,19]. Phosphorylation of IP_5_ by the Ipk1 protein results in inositol hexakisphosphate IP_6_ (phytic acid) [20]. The best-studied IPPs are 5-IP_7_ (IP_7_) and 1,5-IP_8_ (IP_8_). IP_7_ is made by the Kcs1 enzyme family from IP_6_ (Figure 1) [21,22]. Kcs1 can also synthesize PP-IP_4_ from inositol pentakisphosphate IP_5_ [23] (Figure 1). IP_8_ is made by the PPIP5K family by adding a diphosphate group to position 1 of 5-IP7 [24,25,26,27] (Figure 1). PPIP5K members, which are named Vip1 in the budding yeast *S. cerevisiae* and Asp1 in the fission yeast *Schizosaccharomyces pombe*, are bifunctional enzymes with a N-terminal kinase domain and a C-terminal pyrophosphatase domain that converts IP_8_ back to IP_7_ [28] (Figure 1).

The *S. cerevisiae* Ksc1 enzyme is essential for intracellular polyP generation, as a *kcs1∆* (deletion of the *KCS1* gene) strain was identified in a genetic screening as possessing undetectable polyP levels [29]. Interestingly, strains with an *ipk1Δ* null allele, which are also unable to generate IP_7_ (Figure 1) have increased levels of polyP which led to the suggestion that IP_7_ does not control polyP synthesis. Instead, for PP-IP_4_, the other inositol pyrophosphate synthesized by Kcs1 is responsible for polyP synthesis (Figure 1) [29]. A subsequent study characterizing the *ipk1Δkcs1Δ* double mutant strain confirmed that polyP synthesis depends only on the fine control of Kcs1-synthesized IPPs [30]. Conversely, the *S. cerevisiae vip1Δ* strain possesses normal polyP levels, demonstrating that IP_8_ plays no role in the regulation of polyP synthesis [30].

IPPs activate Vtc4 by binding to its N-terminal SPX regulatory domain. The crystal structure of this domain, termed after SYG1/Pho81/XPR1 proteins, reveals a basic surface coordinating the inositol polyphosphates [31]. Extensive binding studies suggested that IP_8_ had a binding affinity for Vtc4 higher than IP_7,_ However, further genetic studies and biochemical assays on purified vacuoles conclusively demonstrated that the IP_7_ synthesized by the *S. cerevisiae* Kcs1 is the physiological ligand activating Vtc4 polyP synthesis [32]. The SPX domain is present in several budding yeast and plant proteins involved in various aspects of phosphate homeostatic control. However, among human proteins only a single protein, the phosphate exported XPR1 [33], possesses this domain. In the fission yeast *S. pombe*, six proteins have a SPX domain including the Pi transmembrane transporter Pet1 and the VTC complex components Vtc2 and Vtc4 (Pombase at www.pombase.org).

To investigate if the Kcs1-controlled regulation of polyP synthesis is evolutionarily conserved, we analyzed modulation of polyP synthesis in another extensively studied yeast model, the fission yeast *Schizosaccharomyces pombe*. *S. pombe* evolutionary roots are considered to go back to early ascomycete lineage and the two yeasts diverged approximately 350 million years ago (reviewed in [34]). Thus, conservation or non-conservation of the control of a biological process between the two yeasts will point to a common or a more diverse control of the specific process in question, respectively.

## 2. Materials and Methods

### 2.1. Strains and Media

*S. cerevisiae* strains used are isogenic to BY4741 (*MATa, his3Δ1; leu2Δ0; met15Δ0; ura3Δ0*) and have been described [30].

*S. pombe* strains used in this study: *vtc4Δ, asp1Δ*, *asp1^D333A^*, *asp1^H397A^, asp1^I808D^*, and *asp1^1−364^* were generated by PCR-based gene targeting using the kanamycin resistance (Kan^r^) cassette as described previously [35]. The genotypes of the strains are listed in Table 1. *Yeast cultures* were grown in rich medium (YE5S) or as indicated in the text.

Plasmids harboring *asp1^365−920^* and *asp1**^365–920/H397A^* are derivatives of pJR2-3XL and expressed under the control of the thiamine-repressible *nmt1^+^* promoter [35,36,37]. To de-repress the *nmt1**^+^* promoter, transformed yeast cells were grown under plasmid-selective conditions in thiamine-less minimal medium (MM) before harvesting.

### 2.2. Poly-P Extraction

Logarithmically growing cultures from the different *S. cerevisiae* or *S. pombe* strains (OD595 = 10 unit) were centrifuged at 1000× *g* for 2 min. The cell pellet was washed in water and resuspended in 250 μL of LETS Buffer (0.1 m LiCl, 10 mm EDTA, 10 mm Tris, pH 8.0, and 0.5% SDS) and mixed with 250 μL of phenol buffered at pH4.8. After adding glass beads the samples were vortexed for 5 min at 4 °C, followed by centrifugation at 15,000× *g* for 5 min at 4 °C. The polyP containing water phase was transferred to a new tube followed by chloroform extraction. The water phase was transferred to a new tube and RNA and polyP were precipitated by adding 2.5 volumes of ethanol and incubating at −20 °C for >16 h. Samples were spun down 15,000× *g* for 10 min at 4 °C, the RNA/polyP pellet was suspended in 100μL of ddH_2_O. RNA concentration was calculated by reading the absorbance at 260 nm.

### 2.3. PAGE Analysis of Extracted polyP

Extracted polyP (20 μg of RNA) were resolved on a 24 by 16 by 0.1 cm gel using a 30% non-denaturing polyacrylamide/bis acrylamide (19:1) in Tris/Borate/EDTA (TBE) buffer. Before loading the samples, gels were pre-run for 30 min at 200 V. Samples were resolved at 5 mA 600 V overnight at 4 °C, until the Orange-G front reached 10 cm from the gel’s bottom. Gels were stained with toluidine blue as previously described [38].

### 2.4. Phosphate Overplus Experiment

The *S. pombe* strain was grown in complete Pombe Glutamate Medium (PMG) [39]. Exponentially growing cells (OD595 = 50–70 units) were centrifuged at 1000× *g* for 2 min washed twice with phosphate-free PMG and divided into five identical cultures (8–10 mL each) in phosphate-free-PMG. After 2 h incubation at 30 °C, buffered potassium phosphate (pH 7.0) was added to a final concentration of 15.5 mM followed by further incubation for 2 h at 30 °C. Cells were harvested at the indicated times, washed with phosphate-free PMG, and stored at −20 °C until polyP extraction.

### 2.5. Quantification of polyP by Malachite Green

PolyP was measured as phosphate after enzymatic digestion of the polymer. RNA/polyP (2–5 μg) were incubated with recombinant Ppx1 and Ddp1 in reaction buffer (20 mM HEPEs pH 6.8; 6 mM MgSO_4_; 1 mM DTT; 100 mM NaCl) for 1 h at 37 °C. Phosphate present in 5–10% of digested RNA/polyP, and undigested RNA/polyP (1 μL) were assayed using the malachite green assay. Samples and phosphate standards were distributed in a 96 well plate and the volume adjusted to 100 μL with ddH_2_O. Then 100 μL of freshly mixed Molybtate (175 mM (NH_4_)_2_MoO_4_; 2 M H_2_SO_4_)/Malachite (0.15 malachite green; 1.4 g polyvinyl alcohol (100,000 MW) in 400 mL H_2_O) solution (4/3) was added to each well. Absorbance was measured at 650 nm after a 10 min incubation at room temperature. Potassium phosphate standard calibration curve was used to determine the concentration of polyP release phosphate after subtracting the amount of phosphate present in undigested samples.

### 2.6. Asp1 Phosphatase In Vitro Assay

The in vitro phosphatase assays were performed using 4 µg of bacterially generated and purified GST-Asp1^365−920^, GST-Asp1^365−920/H397A^ or GST-Ddp1. Before running the assay, IP_7_ was generated via an in vitro kinase assay using the Asp1 kinase as previously described [28]. The GST-tagged proteins were incubated with either IP_7_ or polyP (type 15) in the reaction buffer (150mM Hepes pH 6.8, 250 mM NaCl, 30 mM MgSO4, and 5 mM DTT). All reactions were incubated for 18 h and the resulting inositol polyphosphates resolved on a 35.5% PAGE and stained with Toluidine Blue.

## 3. Results

### 3.1. The S. pombe PPIP5K Family Member Asp1 Controls polyP Generation

The metabolic relationship between inositol pyrophosphates and polyP has been primarily characterized in budding yeast [30]. To determine if polyP regulation by IPPs was a conserved feature, we compared the polyP levels of *S. cerevisiae* wild type, *vtc4Δ*, and *vip1Δ* strains with the respective *S. pombe* strains wild type, *vtc4**∆,* and *asp1Δ*. Kcs1 is an essential protein in *S. pombe* as our tetrad analysis of a diploid, heterogenous *kcs1^+^/ksc1**∆* strain gave no viable *ksc1**∆* spores. Thus, a *S. pombe kcs1Δ* strain could not be included in our analysis. A SPX-containing homolog of the *S. cerevisiae* Vtc4 protein existsin *S. pombe* but has not been investigated (Pombase). To determine if the protein encoded by the *S. pombe vtc4^+^* gene was required for polyP generation, we generated a *vtc4∆* strain for the analysis.

*S. pombe* and *S cerevisiae* strains were grown in rich media YE5S or YPD, respectively, and polyP was extracted from logarithmically growing cultures followed by PAGE analysis of phenol extracted polyP. As published previously, *S. cerevisiae* wild-type and *vip1∆* strains have comparable amounts of polyP (Figure 2, lanes 2 and 3) but polyP is undetectable in the *vtc4∆* strain (Figure 2, lane 4). Likewise, an *S. pombe* wild-type strain has polyP (Figure 2, lane 6), while the *vtc4∆* strain does not (Figure 2, lane 8). Unexpectantly, and in contrast to the *S. cerevisiae vip1∆* strain, deletion of the *asp1^+^* gene encoding the *S. pombe* PPIP5K family member resulted in severely reduced polyP levels (Figure 2, lane 7). As Asp1 is solely responsible for the generation of IP_8_ in this yeast [28], we conclude that polyP generation in *S. pombe* requires IP_8_.

### 3.2. Asp1-Made IP_8_ Modulates polyP in a Dose-Dependent Manner

As a member of the PPIP5K bi-functional enzyme family, Asp1 has a N-terminal kinase domain and a C-terminal pyrophosphatase domain [24]. As the former generates IP_8_, which is used by the latter as a substrate, intracellular IP_8_ levels are regulated by the two opposing Asp1 enzymatic activities. This has allowed the generation of Asp1-variant strains, that have altered intracellular IP_8_ compared to a wild-type strain. All Asp1-variants are expressed in comparable amounts via the native *asp1^+^* promoter as the wild-type *asp1^+^* gene was replaced by the *asp1* variant version(s) in the genome [28,35,40]. *asp1^D333A^* and *asp1∆* strains cannot synthesize IP_8_ (Figure 3c), while strains *asp1^H397A^*, *asp1^I808D^* and *asp1^1−364^* all have approximately two-fold higher IP_8_ levels than the wild-type stain, due to a non-functional or absent pyrophosphatase domain (Figure 3c). Interestingly, strains unable to generate IP_8_ had massively reduced levels of polyP (Figure 3a and quantification in 3b) while strains with higher than wild-type levels of IP_8_ showed a higher than wild-type polyP levels (Figure 3b,c).

Furthermore, wild-type yeast transformants expressing the Asp1 pyrophosphatase domain Asp1^365−920^ on a plasmid, also had reduced polyP levels (Figure 4a). Yeast transformants with high-copy, plasmid-borne expression of *asp1^365−920^* exhibit in severely reduced intracellular IP_8_ compared to control transformants [28]. Reduction in polyP levels was not observed when an enzymatically inactive version of the Asp1 pyrophosphatase *asp1^365−920H397A^* was expressed (Figure 4a). Plasmid-borne expression of this mutant Asp1 pyrophosphatase variant does not alter intracellular IP_8_ levels [28]. Thus, our analysis demonstrates an interrelation between IP_8_ and polyP in the fission yeast *S. pombe* and points to a dose-dependent IP_8_ regulation of polyP generation.

### 3.3. The Asp1 Phosphatase Cannot Metabolize polyP In Vitro

We have shown previously that the DIPP enzyme family homologue Ddp1 from *S. cerevisiae* which hydrolyzes IPPs also has polyP endopolyphosphatase activity [30]. To determine, if this might also be the case for the Asp1 pyrophosphatase, we generated and purified recombinant, bacterially-made GST-Asp1^365−920^, GST *Asp1^365−920H397A^,* and GST-Ddp1 proteins and tested them in an in vitro phosphatase assay using either Asp1-made IP_7_ or PolyP as substrate. As shown in Figure 4b (left panels), Ddp1 and Asp1^365−920^ have phosphatase activity using IP_7_ as the substrate. However, while Ddp1 also has endopolyphosphatase activity, Asp1^365−920^ does not (Figure 4b, right panels). We conclude that the reduction in intracellular polyP upon high-copy expression of the Asp1 pyrophosphatase, is not caused by Asp1^365−920^ polyP hydrolysis but by the reduction of IP_8_.

### 3.4. Polyphosphate Overplus Mechanism Is Conserved in S. pombe

*S. cerevisiae* cells metabolically respond to phosphate starvation utilizing vacuole stored polyP that is rapidly degraded to supply cellular biochemistry with phosphates. The resupply of phosphate to phosphate starved yeast results in a rapid increase in polyP levels far exceeding its normal amount (overplus) [41,42,43]. To study if in *S. pombe* polyP follows the same metabolic fate, a wild type *S. pombe* strain was pre-grown in phosphate-containing minimal medium Pombe Glutamate Medium (PMG). Logarithmically growing yeast cultures were then shifted to phosphate-free PMG medium for 2 h followed by phosphate addition. PAGE analysis of polyP extracted at different time points revealed that shifting *S. pombe* cells into phosphate-free medium caused rapid polyP hydrolysis, resulting in a ~50% polyP decrease (Figure 5a left panel, quantification in Figure 5b). This is in contrast to the >80% decrease recorded for *S. cerevisiae* using a comparable experimental setup [30]. This shallow reduction in polyP observed in fission yeast shifted to phosphate-free condition is agreement with a previous report in which polyP depletion was observed after six hours incubation in phosphate-free medium [44]. Resupply of phosphate induced a rapid synthesis of polyP. One hour after phosphate-resupply, the polyP cellular level exceeded the polyP normally present in exponentially growing yeast (Figure 5a, right panels and Figure 5c). These data demonstrate that polyP in *S. pombe* follows a metabolic fate similar to the one recorded in other microorganisms.

## 4. Discussion

The dual-functionality PPIP5K enzymes generate and fine-tune IP_8_ cellular levels via their N-terminus kinase domain synthesizing IP_8_ and their-C-terminal phosphatase domain that degrades IP_8_ to IP_7_. IP_8_ modulated biological processes are manifold and in the fission yeast *S. pombe* we have shown that these high energy molecules, generated by Asp1, control genome stability by modulating mitotic machineries, adapt the yeast life cycle due to environmental signals and regulate the integrity of the cytoskeleton and mitochondria [28,35,37]. Recently, the Schwer lab has identified Asp1 as a regulator of phosphate homeostasis controlling RNA 3’processing/transcription termination of *S. pombe* phosphate acquisition genes [45]. Our present analysis shows that Asp1 generated IP_8_ also controls polyP synthesis in a dose-dependent manner; thus, pinpointing to a central role of IP_8_ in phosphate homeostasis in this yeast.

Phosphate homeostasis is one of the fundamental biological events regulated by IPPs [46,47]. In budding yeast, this relationship has been investigated from a signaling perspective trough the activation of the PHO pathway [48] as well as metabolically trough the accumulation of polyP inside the vacuole [29,30]. Pioneering works carried out in *S. cerevisiae* elucidated the molecular mechanism regulating polyP vacuolar accumulation: a combination of structural, genetic, and biochemical approaches has emphasized a specific role for the Kcs1 generated 5-IP_7_ in regulating Vtc4 polyP-synthesizing activity, and consequently the accumulation of the polymer inside the budding yeast vacuole [31,32]. Consequently *S. cerevisiae kcs1Δ* strains have undetectable [29] or very low polyP levels [30]. Conversely, *S. cerevisiae vip1Δ* strains unable to synthesize IP_8_ and thus accumulating 5-IP_7_ have normal [30] or even higher than WT levels of polyp [32].

Intriguingly, our present work uncovers that the IPPs selectivity regulating cellular polyP levels is not conserved among species. While the members of the Vtc4 family in both budding and fission yeast are responsible for polyP synthesis, different enzymes in the two species control polyP levels. The *S. pombe asp1Δ* strain possesses IPPs deficiency similar to the budding yeast *vip1Δ* mutant, as it is devoid of IP_8_ and contains higher level of 5-IP_7_ [28]. However, and in stark contrast to the findings for the *S. cerevisiae vip1* strain, the *S. pombe asp1Δ* strain has very low levels of polyP. Analysis of several strains expressing Asp1 variants with deficient kinase or phosphatase function, clearly demonstrated that in *S. pombe* IP_8_ modulates polyP cellular level, and it is doing so in a dose-dependent manner.

How is this distinct IPP regulation of polyP levels in fission and budding yeasts achieved? Crystal structure analysis of the SPX regulatory domain revealed a basic surface where negative charged molecules such as IP_6_ or IPPs can bind [31]. Binding studies carried out on several SPX domains provided an affinity scale whereby IP_8_ > IP_7_ > IP_6_ [31,32] substantially reflecting the decreasing charging nature of the inositol phosphate tested. These structural and binding studies did not reveal a selective binding pocket that may explain SPX IPP discrimination. Therefore, if the differential IPP regulation of polyP cellular level between budding and fission yeast is due to a differential IPP activation of Vtc4, additional structural elements must exist. Potential candidates include another protein or small molecule cofactor that confer selectivity to the SPX domain. A better characterization of the fission yeast VTC complex is necessary to address this important issue. We cannot exclude at present that the different IPPs regulation of polyP level between fission and budding yeast is independent from SPX/Vtc4 paradigm.

The dose-dependent regulation of polyP generation by intracellular IP_8_ levels was manifested by an increase in intracellular IP_8_ levels above wild-type levels leading to a polyP increase above wild-type levels. At present it is unclear how this modulation is achieved on a molecular level. However, it is important to note that Asp1-made IP_8_ also controls microtubule dynamics [37,40]. The *S. pombe* interphase microtubule cytoskeleton controls the nucleus-proximal localization of vacuoles as well as their fusion/fission dynamics [49,50]. As microtubule stability is increased or decreased with increasing or decreasing IP_8_ levels, respectively, it is possible that vacuolar membrane dynamics and thus the membrane bound VTC complexes functionality is influenced by the IP_8_-modulated microtubule dynamics.

## Figures and Tables

**Figure 1 jof-07-00626-f001:**
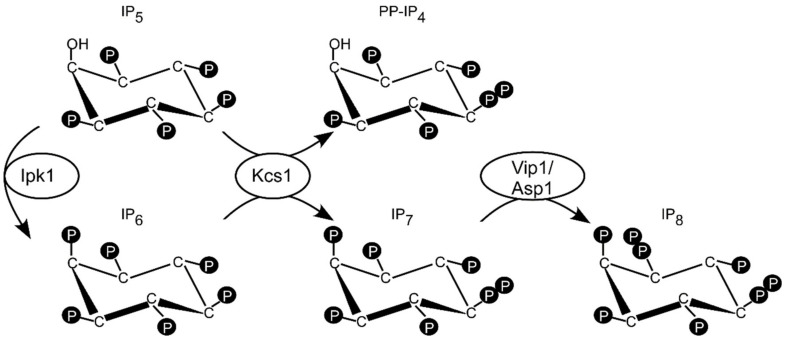
Schematic representation of the inositol pyrophosphates metabolic pathway in *S. pombe* and *S. cerevisiae*. Inositol hexakisphosphate IP_6_ synthesized by Ipk1 is further phosphorylated to IP_7_ by the Kcs1 enzyme, that is also metabolizing IP_5_ to PP-IP_4_. IP_8_ is synthesized by Asp1 (*S. pombe*) or Vip1 (*S. cerevisiae*).

**Figure 2 jof-07-00626-f002:**
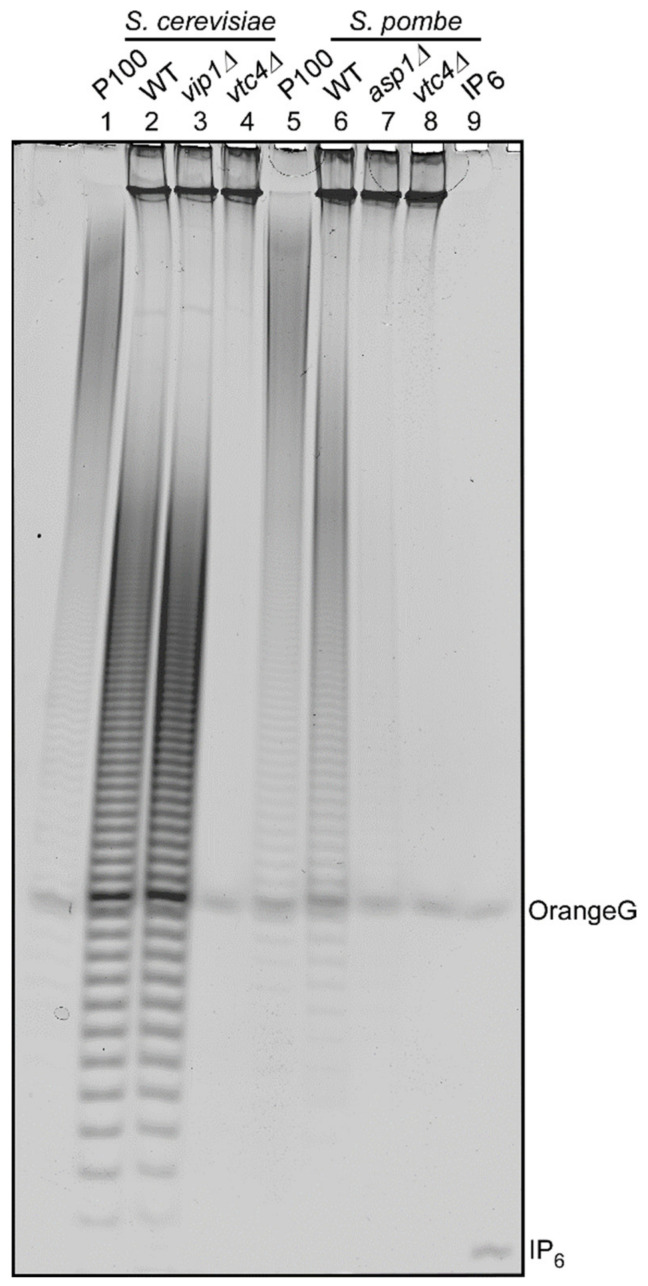
Synthesis of polyP in *S. pombe* depends on Asp1 function. Phenol-extracted PolyP from exponentially growing yeast cultures and equal amounts of RNA (20 μg) were resolved on 30% PAGE and visualized by toluidine staining. The extracts of the following strains were analyzed: (from left to right) *S. cerevisiae* wild-type (WT), *vip1Δ,* and *vtc4Δ* and *S. pombe* wild-type (WT), *asp1Δ,* and *vtc4Δ.* The figure shows a representative analysis from 3 independent experiments with virtually identical results.

**Figure 3 jof-07-00626-f003:**
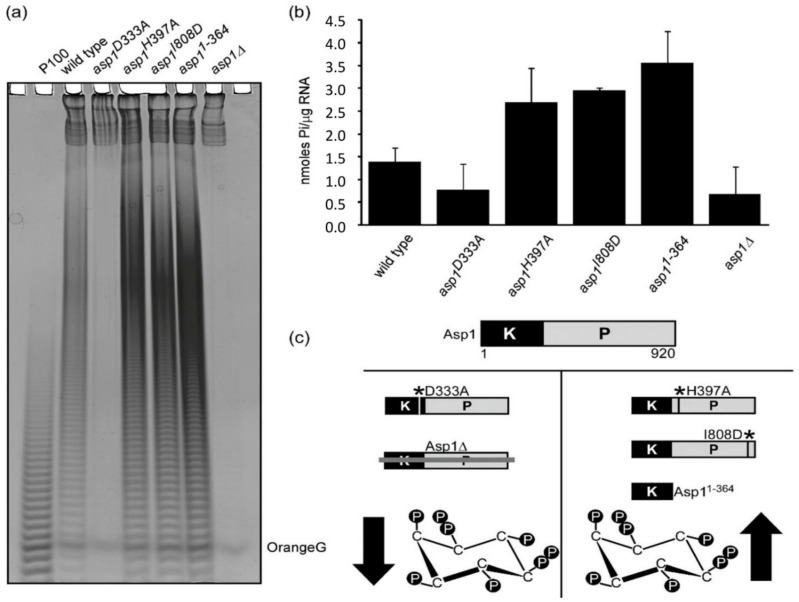
Asp1 generated IP_8_ modulate polyP synthesis in a dose dependent manner. (**a**) Asp1 generated IP_8_ regulate cellular polyP levels. *S. pombe* polyP detected as a dark smear by PAGE analysis of lysates of the wild-type, *asp1*^D333A^, *asp1^H397A^*, *asp1^I808D^*, *asp1^1−364^,* and *asp1Δ* strains. (**b**) quantification of polyP levels shown in (**a**) by Malachite Green assay. (**c**) schematic representation of the correlation between Asp1 pyrophosphatase activity and IP_8_ levels in *S. pombe*. All Asp1 variants are expressed endogenously instead of the WT Asp1. *asp1Δ* and *asp1^D333A^* strains have no IP_8_, while strains *asp1^H397A^, asp1^I808D^,* and *asp1^1−364^* generate more IP_8_ than the wild-type strain. * denotes postion of mutation in Asp1 variant.

**Figure 4 jof-07-00626-f004:**
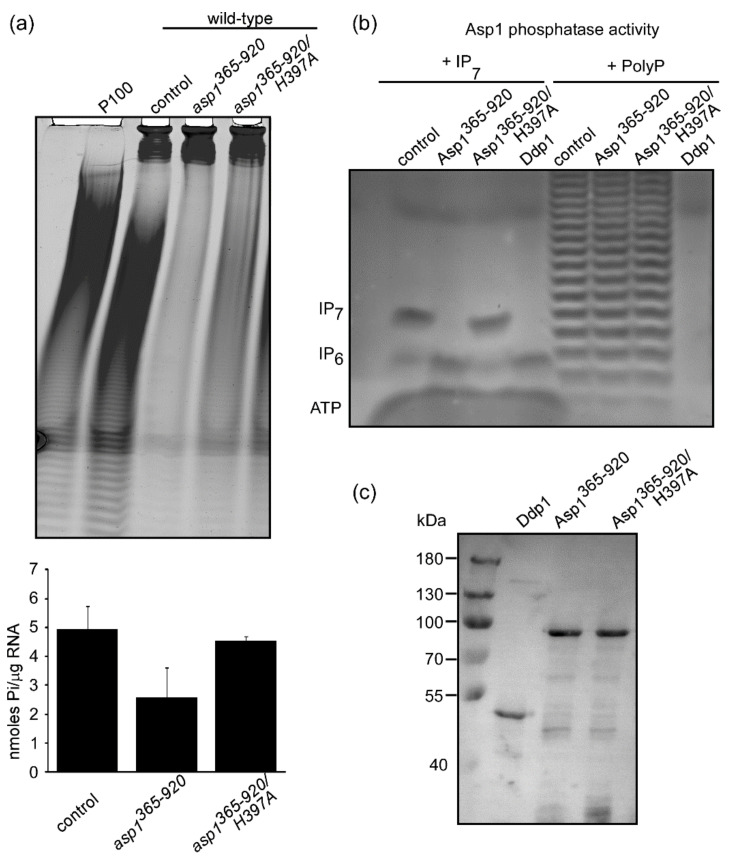
Asp1 phosphatase cannot hydrolyze polyP. (**a**) top panel: polyP analysis (detected as a dark smear by PAGE analysis of lysates) of a wild-type *S. pombe* strain transformed with either a control plasmid or plasmids expressing *asp1^365−920^* or *asp1^365-920/H397A^* under *nmt1^+^* promoter de-repressed conditions. Bottom panel: quantification of polyP levels using a Malachite Green assay. (**b**) the in vitro phosphatase activity of the proteins Asp1^365−920^, Asp1^365−920/H397A^*,* or *S. cerevisiae* Ddp1 were determined using IP_7_ (left panel) or polyP (right panel) as substrate. (**c**) Western blot analysis of GST-tagged proteins Ddp1, Asp1^365−920^*,* and Asp1^365-920/H397A^ expressed and purified from bacteria.

**Figure 5 jof-07-00626-f005:**
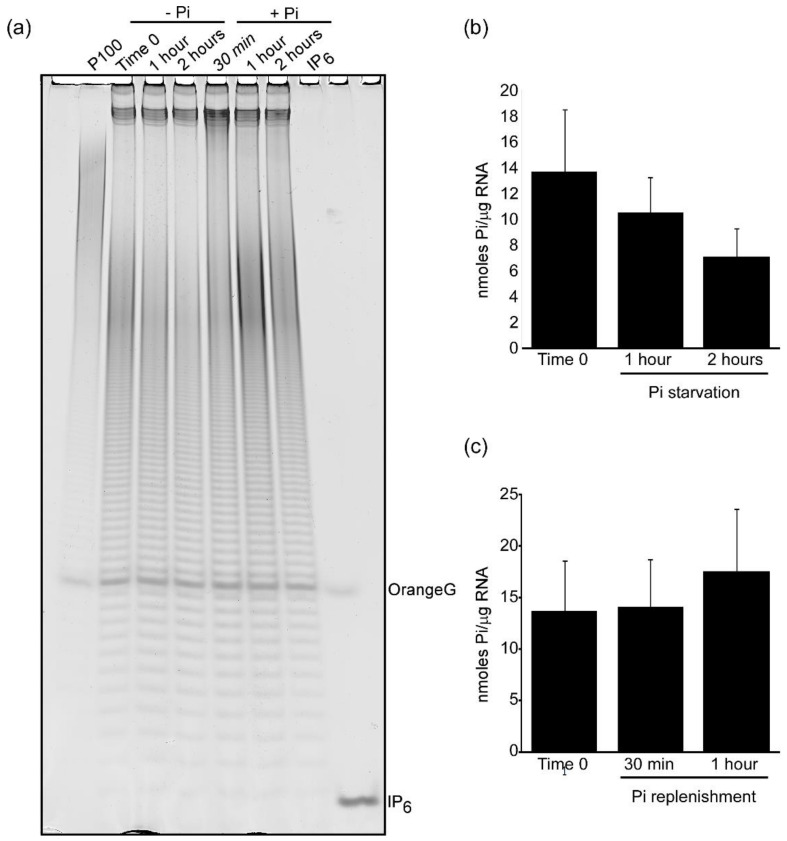
polyP metabolism in response to phosphate starvation is conserved in *S. pombe*. (**a**) polyP profile of *S. pombe* cells during phosphate starvation, -Pi (left) and phosphate resupply conditions, +Pi (right). (**b**) quantification by Malachite Green assay of polyP levels in phosphate-starving cells. (**c**) quantification of polyP levels by Malachite Green assay after phosphate-resupply. The gel shows a representative analysis from four independent experiments quantified as average +/− SD in the graphs.

**Table 1 jof-07-00626-t001:** *S. pombe* strains used in this study.

Name	Genotype	Source
UFY605	*his3-D1, ade6-M210, leu1-32, ura4-D18, h^−^*	K. Gould
UFY1156	*asp1**Δ*::kan*^R^ his3-D1 ade6-M216 leu1-32 ura4-D18, h^−^*	U. Fleig
UFY1511	*asp1^D333A^*::kan*^R^ his3-D1 ade6-M210 leu1-32 ura4-D18, h+*	U. Fleig
UFY1579	*asp1^H397A^*::kan*^R^ his3-D1 ade6-M210 leu1-32 ura4-D18, h+*	U. Fleig
UFY2294	*asp1^1−364^*::kan*^R^ his3-D1 ade6-M21x leu1-32 ura4-D18*, h+	U. Fleig
UFY2553	*asp1^I808D^*::kan*^R^ his3-D1 ade6-M216 leu1-32 ura4-D18, h^−^*	U. Fleig
UFY3033	*vtc4Δ*::kan^R^*his3-D1 ade6-M210 leu1-32 ura4-D18, h-*	U. Fleig

## Data Availability

Not applicable.

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
