# Peer review of "The PPIP5K Family Member Asp1 Controls Inorganic Polyphosphate Metabolism in S. pombe"

_jof, 2021, doi:10.3390/jof7080626_

Round 1

Reviewer 1 Report

The manuscript by Pascual-Ortiz et al, presents interesting observations regarding the mechanism of polyP synthesis and regulation in the fission yeast S. pombe. In general, this is an interesting, well-conducted, and written manuscript. The contribution of this manuscript to the polyphosphate community is considerable, especially considering that little is known about the polyP metabolism in yeast. Consequently, I visualize that this paper will be highly cited alongside the polyP community. 

Minor comments:

  • Line 162: I don't know the policy of MDPI Journals regarding "data not shown". Is my personal view that all results must be shown at least in the supplementary materials.
  •  

Author Response

Line 162: I don't know the policy of MDPI Journals regarding "data not shown". Is my personal view that all results must be shown at least in the supplementary materials.

Answer: We have included a sentence explaining why Kcs1 is an essential protein in S. pombe. “Kcs1 is an essential protein in S. pombe as our tetrad analysis of a diploid, heterogenous kcs1+/ksc1∆ strain gave no viable ksc1∆ spores”.

Reviewer 2 Report

The manuscript by Pascual-Ortiz and colleagues reports the analysis of the molecular mechanism of regulation of polyP synthesis in the fission yeast Schizosaccharomyces pombe, highlighting similarities and differences with the mechanism of regulation in the budding yeast Saccharomyces cerevisiae.

The study design is solid and the manuscript was well written, with the results being clearly reported and discussion and conclusions being correctly derived by the results.

I have only very minor comments on this manuscript, and I believe it is suitable for publication in its current form.

L92: please correct to “the two yeasts”

L163: is “exits” the right word?

L176: I am not sure the authors can tell the generation of polyP is “controlled” by IP8, according to the results shown at this point. Maybe it is more appropriate to say that the generation of polyP is correlated with IP8? (e.g. it is required for the generation, but does not necessarily regulate it).

Author Response

Line 92: please correct to “the two yeasts”

Answer: We have changed it to "the two yeasts"

Line 163: is “exits” the right word?

Answer: We have exchanged it by "exists"

Line 176: I am not sure the authors can tell the generation of polyP is “controlled” by IP8, according to the results shown at this point. Maybe it is more appropriate to say that the generation of polyP is correlated with IP8? (e.g. it is required for the generation, but does not necessarily regulate it).

Answer: We have changed the sentence to:

"As Asp1 is solely responsible for the generation of IP8 in this yeast [28], we conclude that polyP generation in S. pombe requires IP8"